# Research on Product Core Component Acquisition Based on Patent Semantic Network

**DOI:** 10.3390/e24040549

**Published:** 2022-04-14

**Authors:** Wenguang Lin, Xiaodong Liu, Renbin Xiao

**Affiliations:** 1School of Mechanical and Automotive Engineering, Xiamen University of Technology, Xiamen 361024, China; linwg@xmut.edu.cn (W.L.); louis_lxd@163.com (X.L.); 2School of Artificial Intelligence and Automation, Huazhong University of Science & Technology, Wuhan 430074, China

**Keywords:** patent text, core components, big data, structural hole, feature vectors

## Abstract

Patent data contain plenty of valuable information. Recently, the lack of innovative ideas has resulted in some enterprises encountering bottlenecks in product research and development (R&D). Some enterprises point out that they do not have enough comprehension of product components. To improve efficiency of product R&D, this paper introduces natural-language processing (NLP) technology, which includes part-of-speech (POS) tagging and subject–action–object (SAO) classification. Our strategy first extracts patent keywords from products, then applies a complex network to obtain core components based on structural holes and centrality of eigenvector algorism. Finally, we use the example of US shower patents to verify the effectiveness and feasibility of the methodology. As a result, this paper examines the acquisition of core components and how they can help enterprises and designers clarify their R&D ideas and design priorities.

## 1. Introduction

As critical elements of the product, components (i.e., parts), especially core components, are essential for product function [1]. With the rapid development of modern manufacturing, the speed of product iteration is accelerating. For products, the core components determine the performance and quality [2]. More companies are starting to pay attention to the identification and acquisition of core components and to strive for efficient product innovation through research of core components.

Although there are many ways to identify components, companies still lack methods to classify essential core components. As a result, enterprises and R&D institutions cannot fully understand what the core components for a product are, which hinders product R&D. In fact, core components play a vital role in identifying the focus of current product design [3]. However, companies tend to dismantle all the products in the marketplace to identify core parts, which is inefficient and costly.

At present, mining of patent texts is receiving increasing attention from academia. Patent texts are technical and contain nearly 90% of the technical information in the world. Use of available, valid patent information could save companies around 60% R&D time and 40% R&D expense [4]. Patent text has the characteristics of standard format, rigorous language and detailed technical descriptions. In addition, patent text has a sturdy structure, which allows its extraction from modules in order to obtain different information. More importantly, patent application time is crucial for patent text. The combination of patent technology information and application time enables the development of technology for its analyzation and prediction [5].

From the characteristics of the patent text, it can be found that patent text contains rich product information. In order to fully explore product structure information from patent texts and eliminate doubts for product designers, this study proposes a core component identification method based on patent texts, which combines the requirements for core component acquisition and the technical advantages of patent text analysis. Meanwhile, the methodology adopts mining technology of SAO text, which offers extra component description information apart from the patent text. Further, the theories of centrality of eigenvectors and structure holes to analyze the obtained SAO structure are introduced. Finally, the weights of components in products to obtain priority development objectives for enterprises and designers are calculated.

Adopting this methodology enables companies to mine and organize patent text information and identify core components. Compared to dismantling market products, analysis based on patent big data enables analysis of product design in a more comprehensive manner. This helps companies understand which components are critical to product design, analyze the design focus of existing industry, and provide a reference for further research.

## 2. Related Work

### 2.1. Methods of Patent Information Mining

Patent information mining research includes three main methods: patent research method based on (i) international patent classification (IPC) [6], (ii) citation analysis [7] and (iii) patent text [8]. The method based on the IPC, includes the combination of IPC number with the quantity and frequency of related patent numbers. This method reflects the process of technology development and foresees a future development path based on timing characteristics [9]. However, this kind of patent extraction method is relatively rough and based on the macro-level, so it cannot maximize information usage [10]. Patent citation analysis starts from the citation network to analyze the relationship between patents and the process of technology evolution [11]. This method can analyze the development process of technology and future trends. Nonetheless, patent citation analysis cannot ensure timeliness; it also ignores vital information in the contents of the patent [12]. Therefore, the most-used analysis method is based on patent text information mining. In recent years, with the growing number of patents, more scholars prefer using machines for patent text mining [13].

Patent text information mining is mainly divided into three strategies: domain knowledge [14], vector space [15] and patent text semantics [16]. Patent information mining based on domain knowledge comprehensively mines patent technology information by constructing domain knowledge ontology, with the information in the patent text represented by the ontology. However, construction of domain knowledge requires a lot of manual participation, which is time-consuming and labor-intensive [17]. Patent information mining based on vector space models patent information as a vector in order to explore the spatial relationships between vectors, which has the advantage of identifying mature technology [18]. Patent information mining based on the semantic information of the patent text represents the text with semantics as the association rules, allowing for semantic mining and semantic relationship mapping to undertake full mining of the deep patent information, thus it has gradually become the research direction favored in academia [19].

To make the semantic information of patent texts more sufficient, and the technical features of patents to be expressed more clearly, scholars use increasingly efficient methods to mine the semantic information of patent texts, such as entity extraction and SAO. Among them, Ki and Kim [20] used SAO to extract the technical information from a patent by summarizing the defects in existing analysis methods so as to reduce manual operations as much as possible when extracting technical description content, and they displayed the results in the form of a matrix. As for An et al. [21], in order to determine the relationship between keywords in patents, prepositions were introduced into the semantic analysis network, thereby overcoming the limitations of keyword network analysis. Wang et al. [22] considered the relationship between different SAOs when using SAO to mark patent texts and proposed the DWSAO framework to assign different weights to different SAOs, which improves the extraction efficiency of patent text information. In addition to SAO, Chen et al. [23] built a patent information extraction framework through named entity recognition and semantic relation extraction, which improved performance of named entity extraction in patent texts. Xiong et al. [24] monitored, identified and extracted named entities in drug patents, and, by matching knowledge graphs, increased the efficiency of drug information mining in patents to 83%, providing reliable data for building drug monitoring programs. Compared to SAO, the named entity recognition method extracts less deep text information and cannot relate between words and words in patents [25].

### 2.2. Methods of Patent Analyzation

Patent analysis methods are mainly based on four strategies: complex network, vectors, TRIZ and keyword maps.

With the continuous improvement of computer big data analysis and graph computing applications, more academics have begun to emphasize complex networks. High similarity between complex networks and big data in scale, complexity and dynamics resulted in effective correlation in data expression. There are many patent analysis methods based on complex networks, for example, Iwan, Thorsten and Katja [26] took patents as nodes and citation relationships as arcs, showed reference relationships among patents in the form of complex networks and then analyzed the characteristics. Jiang and Zhou [27], through the establishment of complex network and a small-world model, explored the patent performance of industry–university research cooperation; As Park, Chun and Jeong [28] proposed, extracting keywords from patents and analyzing the network in the form of vectors and matrices can identify relationships between keywords and increase patent development efficiency. Choi, Lee and You [29] implemented text mining to extract the characteristics of word vectors, and the document is clustered by an outlier detection algorithm to develop a patent recognition network and explore potential technologies. In addition, Li et al. [30] raised the problem of avoiding patent infringement; this paper puts forward a problem-solving tool based on modern TRIZ tools, which successfully avoids the problem of the patent equivalent of word infringement. Kim M, Park Y and Yoon J [31] built a patent map through patent semantic topic analysis to discover the inheritance relationships between patents. Comparison of patent analysis methods is shown in Table 1:

After comparative analysis, the patent analysis method based on a complex network considers the semantic information in the patent text, can be used for network analysis and has better visualization effect, so it has gradually become a popular method for patent analysis. Since 2019, new ideas of patent mining research based on complex networks have constantly been brought up. It also will be a hot topic in the future [32].

### 2.3. Methods of Core Component Acquisition

For component recognition, identification methods of core components are mainly divided into four categories: network algorithm, machine learning, quality function deployment (QFD) and term frequency–inverse document frequency (TF-IDF). Yin et al. [33] proposed the identification of influential parts of complex mechanical products based on a complex network: the components in the product are mapped to nodes. This method use a local rank algorithm to sort the influence of components. However, it can only analyze a specific product, and the construction of complex networks is quite complicated. For discovering components, Klahn et al. [34] proposed automatic identification of parts by QFD, which is of great significance to design change and technology transfer. Zheng et al. [35] pointed out that components’ healthy and stable operation is the guarantee of stable and safe operation in power system, so they proposed a machine learning method to identify and detect defects in transmission line components. Dorji et al. [36] applied TF-IDF to domain-related terms to identify and extract domain-related terms. The comparison of core component identification methods is shown in Table 2:

It can be seen that core components play an essential role in equipment manufacturing, social progress, and enhancing a country’s hard power. Core components are the most critical part of a product, so identifying core components is of great significance.

### 2.4. Summary

Judging from the research status of component identification, patent information mining and patent analysis methods, existing research mostly focuses on mining product technical information in patent texts to achieve technical efficacy analysis, cluster analysis, etc., but little research has been done in identifying and acquiring core components [37,38,39,40,41,42]. There has been some research on the application of patent text information mining to product functions [43]. In general, current academic circles are paying more and more attention to research of patent text information mining in product development. Therefore, application of patent text analysis has bright future prospects in core component identification and acquisition for both small-scale databases and large-scale databases [44]. In summary, this study proposes to build a semantic network for core components based on patent text big data, integrating the methods of SAO, structure hole and complex network measurement to carry out the identification and acquisition of core components. It is hoped that this method can provide an idea for core component extraction.

## 3. Research Framework

### 3.1. Overview of Core Components

Patented products usually contain multiple components and various connections between them. The importance of components depends on their functions and particular connection frequencies with other components. Based on the previous definition of core components [45], and combined with practical information from the mechanical field, this study makes the following definition:

**Definition** **1.**
*A core component is the component in a crucial position and that occupies more resources in production.*


Resources occupied by a component can be measured by two aspects: (i) the quantity of adjacent components; and (ii) the quality of adjacent components. Suppose the relationship of components in a product as a matrix; the size of the adjacent component matrix depends on the number of adjacent components. Therefore, the nodes with more adjacent components have higher participation in the product, which means they play a more prominent role. In fact, some components are not only adjacent with many other components but also adjacent with other highly important components. Therefore, only considering the quantity but ignoring the quality is not reasonable. Thus, deciding whether a component is essential depends on both the quality and quantity of its adjacent components. For example, the shower in Figure 1 is composed of 10 components. The adjacent matrix shows that components 3 and 4 are connect to the same number of components, but the weight of component 3 (W3 = 8) is larger than that of component 4 (W4 = 7).

Starting from considering the “quantity” and “quality” of the component, adjacent matrixes can identify more-essential components. However, it is difficult to identify those “mediators” between networks because they have fewer adjacent components, so judging whether a component is essential also depends on the component’s location in the product network. To judge whether the position of the component in the product is favorable, first, we start with the number of shortest paths through the component in the product network. A component’s ability as a “mediator” in the product determines its importance. Perhaps there are not many adjacent components for a “mediator”, however, as more components must communicate through the “mediators”, the role of the component becomes more prominent.

It can be considered that once important components are damaged, the whole product system will take a huge loss. The core component is the one in closer contact with others and located in a more-favorable location in the product networks and that undertakes the main functions.

### 3.2. Research Framework of Core Component Acquisition

This study proposes a framework of core component acquisition of patents based on SAO, structure hole and complex network technology. The framework contains four steps, as shown in Figure 2. Firstly, the SAO structure of the patent text is extracted. Secondly, the complex network algorithm and structure hole theory are used to analyze the SAO structure and calculate the component’s importance. Finally, the essential measurement of the core component is obtained according to the calculated data. The steps for acquiring core components are as follows:

Step 1: Patent data retrieval and acquisition

Firstly, retrieve and download industry patent information through websites. Secondly, calculate recall and precision ratio to verify downloaded patent text information will meet the requirements. Finally, select the downloaded patent information from the text.

Step 2: Data preprocessing and supplementation

Check the patent technical information acquired from Step 1 and manually retrieve and supplement the patent text information, avoiding incorrect and garbled patent text information. The format of the patent text is generally divided into five parts: (i) technical field; (ii) background technology; (iii) invention content; (IV) description of the drawings; and (v) specific implementation. The background technology part mainly introduces the background of the patent, the technical background and the significance of the applied patent. The invention content part mainly explains the applied patent in detail. The diagram of the patented product is explained in the illustration part. Finally, the specific implementation part states the content that the patent wants to protect. 

The patent overview includes the description of the patent or the technical points of the utility model; this method extracts the invention part from patent text. Then, remove the label in the text by using a regular expression. After that, manually delete other interference information in the text. Finally, after preprocessing and supplementing the patent overview information, regular expressions are used to segment the text information.

Step 3: Part-of-speech tagging and SAO acquisition

The SAO structure will be extracted from POS tagging obtained in Step 2. Since there are varieties of synonyms in the patent text, it is inaccurate to statistically analyze the SAO structure extracted in Step 2 directly. Use sememe similarity to compare and merge words with the same meaning. Then, screen and eliminate the nodes of unrelated components in SAO.

Step 4: Complex network construction and core component acquisition

Prepare for construction of the SAO complex network by calculating the node’s weight according to the measurement value of S (subject) and O (object) in the structure of SAO. After processing the SAO structure, take “S” as the source node, take “A” as the edge and “O” as the target node to form a directed graph. Finally, apply the structure hole algorithm and eigenvector centrality to calculate the obtained SAO to get the core components. 

## 4. Methods

### 4.1. POS Tagging and SAO

POS tagging is a technology used in natural language processing (NLP), also known as grammar tagging. Valuable information in the text can be extracted by tagging the POS in the text. The primary forms of POS in the text are shown in Table 3: 

Subject–action–object (SAO) or subject–predicate–object is triple extracted from text. Among SAO, the subject and object usually appear as nouns, representing the executor and the executed, respectively, in the event. The predicate connects the subject and object as action. A sentence may contain only one group or multiple groups of SAO structures. In patent text, the structure of SAO in reference Guo et al. [43] is summarized as shown in Table 4:

Components appear with high frequency in patent texts and mainly exist as subjects and objects in SAO structure, while the connection between components is mainly based on function as a “bridge”. Taking “fast block valve” as an example, the SAO structure connecting components with functions is shown in Table 5:

Applying SAO to an information-dense text can quickly and concisely explain the central idea of the sentence, reducing the workload in reading and analyzing. Furthermore, SAO ignores unimportant information in the text but quickly understands the core content. SAO is widely used in natural language processing, text information mining and artificial intelligence. Although the general keyword extraction method can extract keywords, it ignores the relationships between them, which reduces the utilization rate of data. Therefore, using SAO extraction technology does not only extract the main information in the text, but also retain the relationships between the key information. Component information and technical information can be collected accurately and quickly by extracting the SAO structure from the patent text, and the Algorithm 1 of SAO extraction process is described as follows:
**Algorithm 1:** The algorithm of SAO extraction process.*01.//Variable: PT——patent text**02.//Variable: ST——sentence**03.//Variable: WD——word;**04.//Variable: WDPOS——word with POS tagging**05.//Function: SAO——subject-action-object**06.//Function: POST——POS tagging**07.//Function: DEP_GRAM——dependency grammar**08.//Variable: DataFrame——distributed data set**09. Begin:**10. import nlp package**11. import CSV to DataFrame**12. Subject_POS_list = [“Subj”, “nsubj”, “nsubjpass”]**13. Action_POS_list = [“aux”, “auxpass”, “complm”, “prt”, “cordmod”, “mmod”]**14. Object_POS_list = [“obj”,”dobj”, “pobj”]**15. **def** getSubject (ST):**16.   **for** word in ST:**17.    WD = word**18.    WDPOS =POST(word)**19.    If WDPOS in Subject_POS_list:**20.     Subject = WD**21.   Return Action**22. **def** getAction (ST):**23.   **for** word in ST:**24.    WD = word**25.    WDPOS =POST(word)**26.    If WDPOS in Action_POS_list:**27.     Action = WD**28.   Return Action**29. **def** getObjection (ST):**30.   for word in ST:**31.    WD = word**32.    WDPOS = POST(word)**33.    If WDPOS in Object_POS_list:**34.     Object = WD**35.   Return Object**36. data = pd. DataFrame()**37. **for** i in range(len(csv)):**38.   ST = split (PT, Regular expression clause)**39.   **for** j in range (len (ST)):**40.    Subject = getSubject (WD)**41.    Action = getAction (WD)**42.    Object = getObject (WD)**43.    SAO = (Subject,Action,Object)**44.   end**45.   data = data.append(SAO)**46 **end for**.**47. output(data))*

### 4.2. Semantic Similarity

There may be one or more interpretations for the same word. For example, the word ‘handle’ has two interpretations as a noun, namely (a) the part by which a thing is held, carried, or controlled; and (b) a name or nickel. As a verb, ‘handle’ also has two meanings: (a) feel or regulate with the hands; and (b) manage (a situation or problem). According to the meaning ‘feel or regulate with the hands’ as a verb, its synonyms are ‘hold’, ‘pick up’, ‘grab’, ‘grip’, ‘lift’, ‘feel’, ‘touch’, ‘finger’, ‘thumb’, ‘touch with’, ‘play with’ and ‘paw’. According to its meaning as a noun, ‘the part by which a thing is held, carried or controlled’, similar words are ‘have’, ‘shank’, ‘stock’, ‘shift’, ‘grip’, ‘handgrip’, ‘hilt’, ‘helve’ and ‘but knob’. Moreover, take the word ‘hold’ as an example—as a verb, it can be interpreted as grab, carry, or support with one’s hands; as a noun, it means an act or manner of grasping something, and it also means power or control. These meanings are consistent with the interpretation of ‘handle’. Therefore, the similarity of the interpretation of two words determines the similarity between them. 

Suppose that the degree of similarity between a word and itself is 1; the similarity between two words with completely different meanings is 0. Set the word $w_{i}$ to contain sememes set $\left(w_{i1},w_{i2},\cdots ,w_{im}\right)$, the word *w_j_* has interpretations set $\left(w_{j1},w_{j2},\cdots ,w_{jm}\right)$, then the formula of semantic similarity (*SS*) between word $w_{i}$ and word $w_{j}$ is as Formula (1):(1)$${SS}_{\left(w_{i},w_{j}\right)}=\underset{\begin{array}{l} 0\leq p\leq m\\ 0\leq q\leq n \end{array}}{\max}\left|SOI\left(w_{i}{}_{p},w_{jq}\right)\right|$$
where *SOI*(*w_ip_*,*w_jq_*) means similarity of interpretation. In order to calculate semantic similarity accurately, the definitions of words are divided into four characteristics: (1) basic definition description; (2) other definition description; (3) relational definition description; and (4) relational symbol description. Due to the different features of word interpretation, the influence of interpretation features on semantic similarity is also diverse. Thus, the weight of interpretation in semantic similarity is different. Based on this consideration, the weight factor β is added to the calculation of semantic similarity. Then, calculation of semantic similarity is shown in Formula (2):(2)$${SS}_{\left(w_{1},w_{2}\right)}=\sum_{i=1}^{4}\beta_{i}\underset{\begin{array}{l} 0\leq p\leq m\\ 0\leq q\leq n \end{array}}{\max}\left|SOI\left(w_{i}{}_{p},w_{jq}\right)\right|$$
where $\beta_{1}$ is the basic interpretation of the word, $\beta_{2}$ represents other definitions, $\beta_{3}$ is the relationship interpretation and $\beta_{4}$ is the relationship symbol. Further, $\beta_{1}$ + $\beta_{2}$ + $\beta_{3}$ + $\beta_{4}$ = 1, and $\beta_{1}$ > $\beta_{2}$ > $\beta_{3}$ > $\beta_{4}$. Since $\beta_{1}$ is the most apparent basic interpretation, the weight of $\beta_{1}$ is defined at above 0.5 according to the research of [46].

Sememe is the smallest unit of interpretation, and a finite set of sememes can express an interpretation. When calculating sememe similarity, it is usually calculated by the path length between two sememes in the sememe tree. When two sememes are on different sememe trees, the path length between the two sememe nodes is considered infinite, which means the similarity between them is 0. To calculate the interpretation similarity, Formula (3) can be applied:(3)$$SOI\left(w_{i}{}_{p},w_{jq}\right)=\frac{\alpha }{dst(s_{i1},s_{i2})+\alpha }$$

The $dst(s_{i1},s_{i2})$ in Formula (3) is the path distance between two sememe nodes, and α is the adjustable parameter. The sememe tree model is shown in Figure 3:

Taking the sememe tree in Figure 2 as an example, “Root” is the root node, and other nodes can find at least one path to reach the root node. In addition, *B*_1_ is the root node of *B*_11_, *B*_12_, and so on. When two nodes do not have a common root node, it means that the two nodes have no path to reach each other, so it can be regarded that the meanings expressed by the two sememes are not similar.

### 4.3. Complex Network

The term “semantic network” was proposed by Quillian [47]. It is a knowledge base with a directed graph structure in the knowledge graph. It is mainly used for understanding natural language. A semantic network can describe the state and the relationship between things.

The directed graph of a semantic network is composed of nodes and arcs (i.e., edges) between nodes. Nodes are mainly used to represent events or objects, and arcs represent the relationships between them. For example, data from Table 5—taking some components as nodes, the functions between components as arcs, and the degree as the node size—can be used to construct a semantic network model, shown in Figure 4:

Apparently, a semantic network has the advantage of expressing the relationship between nodes directly and clearly. Moreover, a semantic network can efficiently analyze and infer the relationships between nodes. With their continued development, more concepts and theories based on semantic networks have been applied to politics, the economy and intelligence analysis. Using semantic network to visualize the SAO structure in patent text, combined with the concepts and methods of structure hole and feature vector centrality in semantic networks, the information in patent text can be fully mined and displayed.

#### 4.3.1. Calculation of Element Position Importance Based on Structural Hole Theory

The structural hole theory was first proposed by Ronald Burt in 1992 [48]. A structural hole refers to a social phenomenon that a gap exists between individuals in society due to non-direct connection. This phenomenon is named as structural hole, as a hole seems to appear on the whole network. According to this theory, the position of an individual in the network is more important than the strength of the personal relationship. When the structural hole occurs between two network nodes, the node between them without direct connection has stronger information and control advantages. With these advantages, the node can obtain more resources and value. As shown in Figure 5, there is no direct connection from node A, B or C to E, but node D links them. According to Burt’s structural hole theory, node D obtains more information because it occupies a structural hole. Therefore, the more structural holes a node occupies in the network, the more external connections it obtains, and it becomes more favorable.

According to the structural constraint algorithm, the higher the network constraint coefficient is, the better the network closure, and the fewer structural holes exist in the network. The calculation of the network constraint coefficient is shown in Formula (4):(4)$$stren_{ij}=\frac{\alpha_{ij}+\alpha_{ji}}{\sum_{k}\left(\alpha_{ik}+\alpha_{ki}\right)}$$
where $\alpha_{ij}$ and $\alpha_{ji}$ represent the weight of the connected edges between nodes; $\alpha_{ik}$ and $\alpha_{ki}$ represent the total weight of the connected edges between node *i* and all nodes; ${stren}_{ij}$ represents the strength of the connection between node *i* and node *j*.

From the strength of the connection between nodes, the network constraint coefficient (NC) of node *i* can be calculated by Formula (5):(5)$$NC_{i}={\sum_{j}\left(stren_{ij}+\sum_{k}stren_{ik}\cdot stren_{ki}\right)}^{2}$$

The value of k is not equal to *i* and *j*. It can be seen from the above formula that when *j* is the only connected node of *i*, $NC_{i}$ achieves its largest value, which is 1. When j has no direct or indirect connection with node *i*, $NC_{i}=stren_{ij}$. From the calculation results of the network constraint coefficient, the critical node is the one with a small constraint coefficient. From the structure constraint algorithm of structure hole, we can see that the smaller the constraint coefficient is, the more structure holes it occupies in the network, and the more information it obtains.

Figure 5 shows that node D is the only way for other nodes to contact node E. According to the intermediary centrality algorithm in the structural hole theory, the two non-direct contact nodes that get contact through other nodes will be controlled and restricted by the nodes on the path. Consider the product as a network and the components as nodes, suppose the shortest path number from component *q* to component *j* is *g_qi_*, the number of shortest paths from element q to element *j* passing through element *i* is $n_{qj}^{i}$, then the betweenness centrality calculation of node *i* is shown in Formula (6):(6)$$BC_{i}=\sum_{i\neq q\neq j}\frac{n_{qj}^{i}}{g_{qi}}$$

Taking the situation in Figure 5 as an example, the shortest paths between non-directly connected nodes are: A-B-D-E, A-D-E, A-C-D-E, B-D-C, B-A-C, B-D-E, C-D-E. That is, $g_{AE}=3$, $g_{BC}=2$, $g_{BE}=1$, $g_{CE}=1$, the shortest paths from node A to node E all pass through D, then $n_{AE}^{D}=3$, and only one shortest path from node B to node C passes through node D, so $n_{BC}^{D}=1$, and so on, $n_{BE}^{D}=1$, $n_{CE}^{D}=1$. Finally, the betweenness centrality of the node D is $BC_{D}=3/3+1/2+1/1+1/1=3.5$.

The formula of component position importance (CPI) in node *i* network can be obtained by combining the network constraint coefficient and the intermediary centrality algorithm in the structure hole, as shown in Formula (7):(7)$$CPI_{i}=\left(1-\sigma \right)BC_{i}+\sigma (1-NC_{i})$$
where σ is the coefficient of $BC_{i}$, and the value is set to 0.5 according to experience.

#### 4.3.2. Calculation of Component Resource Occupancy Based on Eigenvector Centrality

Academia prefers to use the concept of eigenvector centrality to measure the importance of a node in a network. Eigenvector centrality (EC) not only considers the degree of the node itself but also considers the importance of the node and its connected nodes.

Suppose a product is a network of components, and components are nodes of the product network. Based on the degree centrality of nodes, a product has n components, and take the component library as $\left(e_{1},e_{2},\cdots ,e_{n}\right)$, $\alpha_{\left(i,j\right)}$ is the relation component matrix of *e_i_*, then the calculation method of component metrics (*CM*) of *e_i_* is shown in Formula (8):(8)$$CM_{i}=\sum_{j=1,j\neq i}^{n}\alpha_{\left(i,j\right)}$$

From Formula (8), although the degree centrality can identify nodes with higher connectivity to other nodes in the product network, it only considers the degree centrality and ignores the importance of associated nodes. According to the centrality of eigenvectors, the importance of a node depends on both the quantity and the quality of its adjacent nodes. For example, in Figure 6, B, D and E have a higher I/O degree than the others. Thus, B, D and E occupy more resources. However, F and G have to pass through E to get in touch with other nodes, so node E holds a relatively higher resource amount than F and G; therefore, E is more critical than B and D.

Suppose the relative node matrix of *e_i_* is $\alpha_{\left(i,j\right)}$; the importance measure of *e_i_* is *v_i_*. The calculation method of component resource occupancy (CRO) of *e_i_* is shown in Formula (9):(9)$$CRO_{i}=c\sum_{j=1,j\neq i}^{n}\alpha_{\left(i,j\right)}v_{i}$$
where *c* is a constant. Since *CRO_i_* equals *v_i_*. suppose $V={\left[v_{1},v_{2},\cdots ,v_{n}\right]}^{T}$. A is the component’s iteration matrix of the associated components. After several iterations, V=cAV and the constant *c* is the inverse of the eigenvalue of matrix A.

The centrality of eigenvector is based on the centrality of degree, combined with measurement of the importance between nodes. This enables accurate identification of the core components. Therefore, the core components can be obtained by comprehensively considering the “quantity” and “quality” of component association.

#### 4.3.3. Comprehensive Weight Calculation of Core Components

Based on the core component definition and identification methods mentioned in the former section, to calculate component importance (CI) for component *i* (i.e., *CI_i_*), Formula (10) is applied:(10)$$CI_{i}=\alpha CPI_{i}+(1-\alpha )CRO_{i}$$
where α is the parameter of $CPI_{i}$, specified as 0.5; CI is the value between the interval [0,1]. Using Formula (10) as an indicator, the higher the CI of the component, the more important it is.

In previous research, the identification and acquisition of component importance had not considered the component’s location and associations in the product. Formula (10) can make up for the shortcomings of existing research. CPI is based on the location of components, and CRO is based on the resources (associations) owned by components. Through parameter adjustment, component importance data under different focuses can be obtained.

## 5. Illustration and Discussion

### 5.1. Research Background

As a typical product in the field of mechanical design, a shower is a rapidly updating and high demand product. Due to fierce competition in the marketplace, enterprises upgrade their products to meet the requirements of mainstream consumer groups (post-1980s and 1990s consumers) by adding concepts such as energy conservation, environmental protection and intelligence. However, many companies do not understand the core components and main functions of shower products, which results in obstacles in product design and upgrading. The main reason for the low efficiency of transforming design schemes into market products is insufficient understanding of the product’s core components, in other words, the core components of the designer’s supervisory consciousness have deviated from the facts. Based on these problems, this study takes US shower patents as the research object to obtain the shower’s core components.

### 5.2. Retrieval and Acquisition of Patent Data

The cooperative patent classification (CPC) is more detailed than the international patent classification (IPC). Therefore, a US shower patent was chosen as the research object. This case searched for and downloaded US shower patent text prior to 2020 through the well-known patent website USPTO database (http://www.uspto.gov accessed on 1 December 2019). A total of 1733 US shower patent texts were obtained with the keywords and CPC numbers in Table 6, in which the * means string of character of any length. The results met the requirements of patent analysis by conforming to the recall ratio and the accuracy ratio.

### 5.3. Data Preprocessing and Supplementation

First, we checked the shower patent texts and queried and supplemented missing shower patent text manually. Next, we extracted the patent overview section of the text and applied the BeautifulSoup module in Python to remove the label section, the title section and the garbled section from the text.

### 5.4. POS Tagging and SAO Acquisition

There are three kinds of NLP libraries currently: spacy, NLTK and core NLP. Because Spacy has certain advantages in terms of application ability, running speed and accuracy, it was used for extracting SAO structure and to annotate the POS of words in the text. The first step was to put the overview section of the obtained patent text into a CSV file, then, the overview was read line by line, and the punctuation in each sentence was divided into sentences in the form of regular expressions. Finally, all the sentences were listed, and Spacy was used to mark the POS of the words. Next, we extracted the subject, action, and object in the text according to POS tagging. Some sentences contained multiple subjects, actions, and object keywords. After matching, a total of 1048574 SAO structure groups were acquired. Each SAO structure group was input into the DataFrame according to patent number. Finally, the DataFrame was saved locally in CSV format. The results are shown in Table 7:

In order to verify the effectiveness of this method, we compared our results with TF (term frequency) and TF-IDF keyword extraction algorithms, which are traditional methods [37,49,50], by precision (P), recall (R) rate and F-measure (F) Equations (11)–(13).
(11)$$P=\frac{TP}{TP+FP}$$
(12)$$R=\frac{TP}{TP+FN}$$
(13)$$F=\frac{2\times P\times R}{P+R}$$

Ten patents that contained more than 600 words were randomly selected as test objects, and experts were invited to verify the effect. The results are shown in Figure 7. The ROC value of our technique (SAO-complex network (SAO-CN)) is significantly better than that of other algorithms.

There are some words with high similarity in the combination of SAO keywords, such as ‘head’ and ‘heads’; ‘contained’ and ‘containing’. Generally, the similarity between words and their plural forms and the primary meaning of passive voice is 1. Thus, synonyms are combined for nodes with the original similarity of 1. The degree of a node is taken as the weight to measure the size of the node. After statistical treatment, the weight of the corresponding nodes are shown in Table 8:

### 5.5. Complex Network Construction and Core Component Acquisition

Gephi is software for analyzing complex networks. In this study, Gephi was applied to analyze the shower patent case. For analyzing the shower patent text and building a complex network: firstly, import the CSV file into Gephi; take ‘subject’ as the initial node, ‘label’ as the directed edge, and point to the node target at ‘object’. Finally, 3099 groups of component nodes are obtained by selecting and eliminating the SAO structures independent of components. Screen the core nodes by the K-core algorithm, and obtain the complex network of SAO keyword combinations of shower patent text, as shown in Figure 8:

According to Formula (9), the occupancy of component resources (CRO) is obtained by taking the centrality of eigenvector as the judgment standard, as shown in Table 9.

According to Formula (7), the position importance of components (CPI) is obtained from the betweenness centrality and structural constraint coefficient of components (the calculation results shown in Table 8). Because the intermediate centrality coefficient value is small, the coefficient is adjusted to one decimal place. According to the values listed in Table 7 and Table 8, combined with Formula (10), component importance is calculated as shown in Table 9. Due to the nature of this study, only the top 20 components are listed here.

### 5.6. Demonstration and Discussion

Furthermore, we selected US patent applications from 1 January 2020 to 31 December 2021 as a test dataset, which included 72 patents and 30,000 words, to evaluate the predictive ability of the method, and we used P, R and F calculated by Equations (11)–(13) as ability indexes. By importing the data in Table 9 into a test dataset, the scores of P, R and F were 0.75, 0.79 and 0.77, respectively. Those high scores mean that the performance of the proposed method is reliable. After all, it is hard to detach a few valuable components from a huge amount of patent data for R&D.

As shown in Table 9, patent data mining determined a valve is the core component of showers. In fact, the valve of showers in the market is mainly divided into single-way, double-way or three-way. Showers with more complicated valves can accommodate more water flow patterns, meeting diverse customer needs. In light of this, we designed a new shower with a motor-controlled valve that enables multiple types of water flow, as shown in Figure 9. The working principle of this shower is: when the water has passed through coil 5, the coil will generate electricity, which will be stored in the battery placed inside the showerhead 25; the battery will be used to support the motor 14 to adjust the valve 15, thereby allowing water flow control. The motor can be controlled by a microchip (not shown). Compared to existing products, the valve has a simpler mechanical structure and automatic-adjust function by using the motor and electric circuit.

1. inlet joint; 2. handle; 3. magnet impeller; 4. spacer casing; 5. coil; 6. sealing cover; 7. O-ring; 8. reversing stud switch; 9. screw; 10. sealing cover; 11. irregular shape ring; 12. cavity; 13. limit block; 14. motor; 15. valve; 16. long screw; 17. surface cover; 18. waterproof sealant; 19. shaft; 20. limit block; 21. gasket; 22. water distributor connector; 23. screw; 24. decorative cover; 25. showerhead.

This section introduces a case for the method proposed in this paper. Computational results and algorithm comparison results fully demonstrated the feasibility and effectiveness of this method, from which it can be inferred that exciting results will be obtained by applying this method to emerging and popular mechanical products.

## 6. Conclusions

The manufacturing industry is the main body of the national economy. In the new round of industrial reform and the remodeling of the international industrial division pattern, product manufacturing enterprises should attach importance to core components in product design and manufacturing, enhancing the competitiveness of products and improving the efficiency of product R&D. The main findings of this study are:

(1) Combined with existing data, we identified the shortcomings of existing research on core components. This study proposes critical information mining in patent text to improve the accuracy of core component acquisition.

(2) To fully mine patent text information, this study proposes a framework of core component acquisition integrating SAO and structural holes. The method includes five steps: 1. patent data retrieval and acquisition; 2. data preprocessing and supplementation; 3. SAO structure acquisition; 4. synonym merging and data acquisition; and 5. complex network construction and core component acquisition.

(3) Based on the method and framework mentioned above, critical technical issues such as extraction of the SAO structure by part-of-speech tagging, the combination of synonyms by sememe similarity, and analysis and judgment of the extracted SAO structure by using structural holes theory and eigenvector centrality were elaborated on.

This study applied a multi-method perspective, combining patent big data, natural language processing, social science and a complex network. We expound upon and discussed core component acquisition of shower patents in the United States to verify the method’s feasibility.

## Figures and Tables

**Figure 1 entropy-24-00549-f001:**
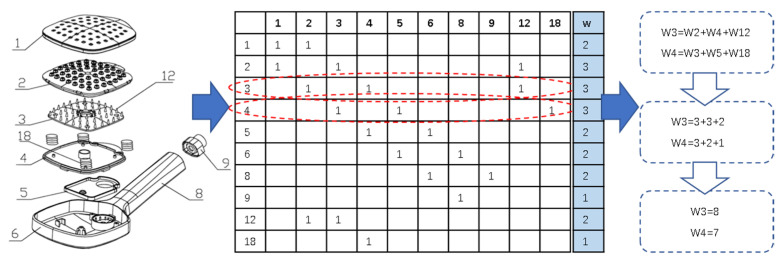
Example of core component.

**Figure 2 entropy-24-00549-f002:**
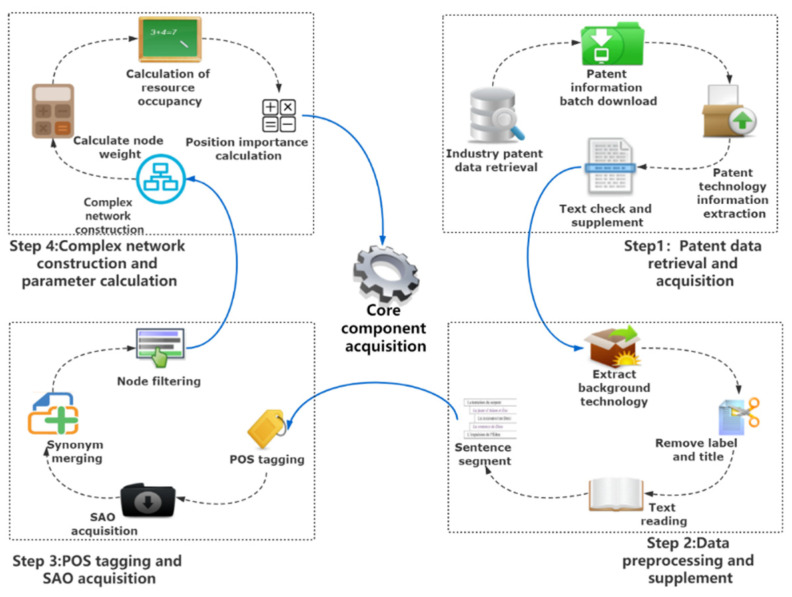
Framework of core component acquisition.

**Figure 3 entropy-24-00549-f003:**
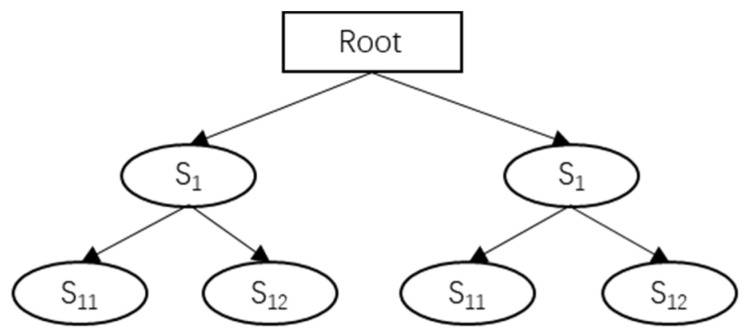
Semaphore tree model.

**Figure 4 entropy-24-00549-f004:**
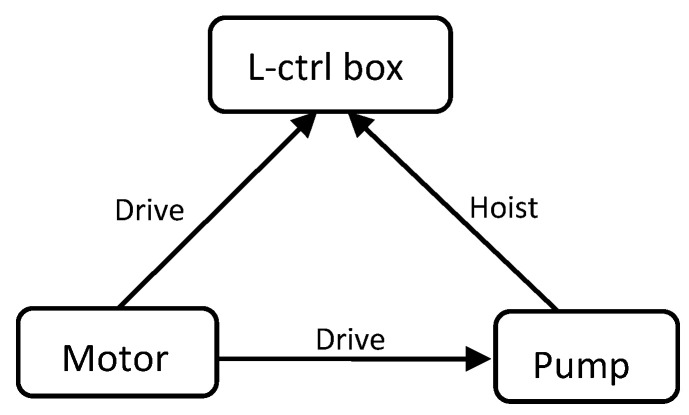
Semantic network model.

**Figure 5 entropy-24-00549-f005:**
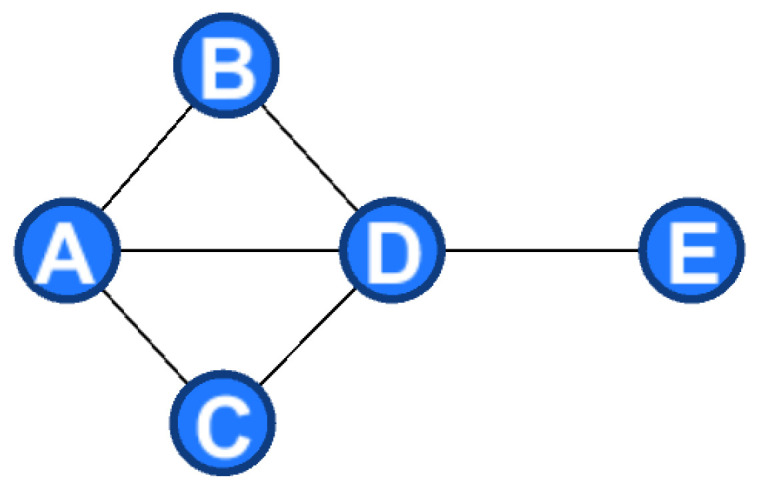
Example of structural hole network.

**Figure 6 entropy-24-00549-f006:**
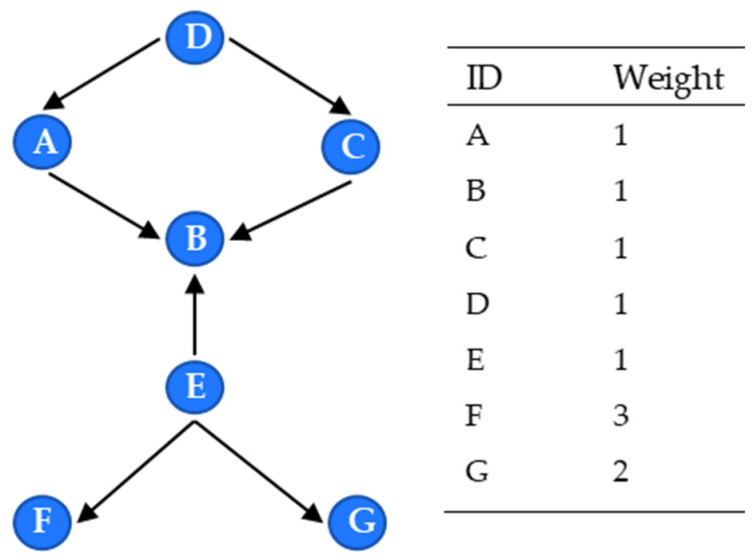
Example of eigenvector network.

**Figure 7 entropy-24-00549-f007:**
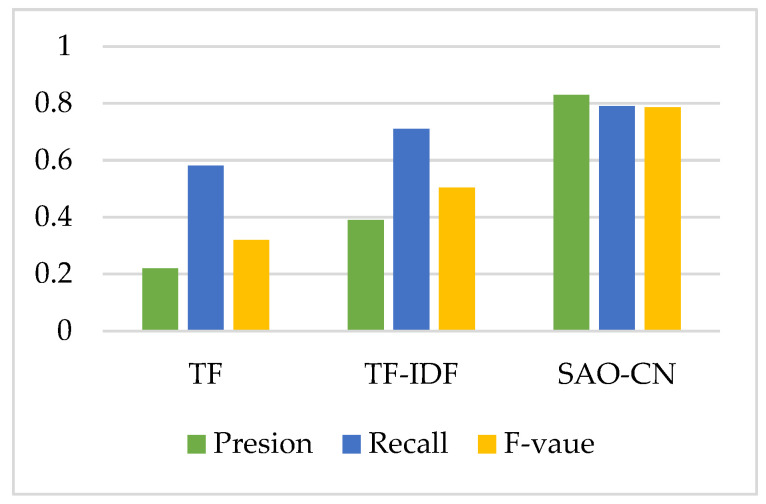
Comparison of three algorithms.

**Figure 8 entropy-24-00549-f008:**
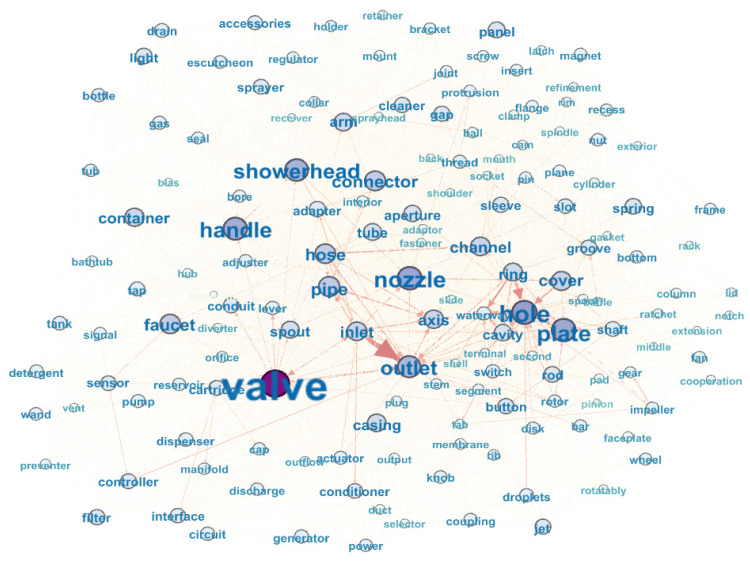
Complex network of SAO keyword combinations for shower patents.

**Figure 9 entropy-24-00549-f009:**
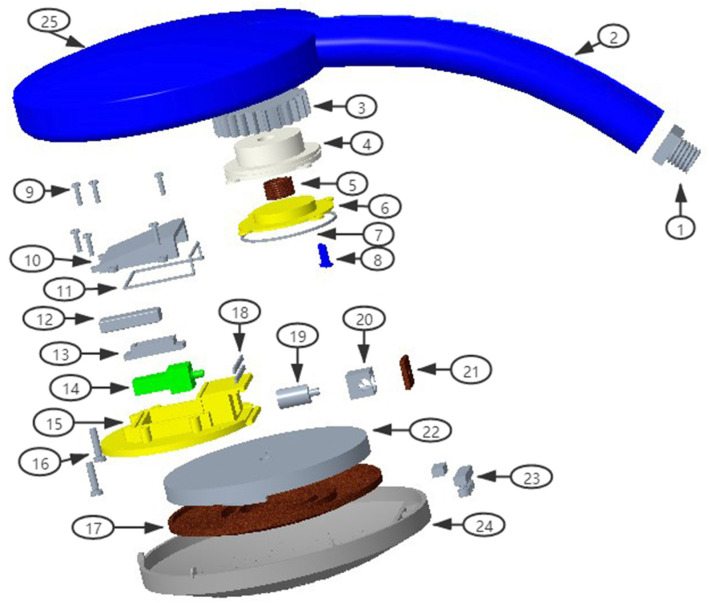
Shower product structure.

**Table 1 entropy-24-00549-t001:** Comparison of patent analysis methods.

Method	Mode of Action	Advantage	Disadvantage	Literature
Complex network	The keywords in the patent text are regarded as nodes, the associations between keywords are regarded as edges, and the complex patent network is constructed for analysis	Strong visualization, which is conducive to clarifying the relationship between keywords and facilitating network analysis	Insufficient dynamic visualization	Iwan, Thorsten and Katja
Vector	Performs word vector training on the domain corpus to construct technical efficacy topics	Suitable for large databases, high degree of automation	Lack of judgment on the semantic connection of keywords	Park, Chun and Jeong
TRIZ	Through the analysis and extraction of patent knowledge, it is introduced into TRIZ tool to provide a large number of heuristic principles, effects, structures, etc. for solving product innovation problems in specific fields.	Not only makes up for the limitations of TRIZ and the ambiguity and broadness of the obtained solutions, but also makes up for the microscopic nature of knowledge acquired through patents.	Relies on the designer’s subjective experience and domain knowledge	Li et al.
keyword map	Transforms technical information in patents into a map of technology-directed functionality	The content is detailed and helpful for understanding technological trends	Difficult to find and organize information.	Kim M, Park Y and Yoon J

**Table 2 entropy-24-00549-t002:** Comparison of core component identification methods.

Method	Mode of Action	Advantage	Disadvantage	Literature
Network algorithm	Map product components to networks, and identify core components through network measurement algorithms.	Easy to measure position and use of nodes in network, strong visualization.	Insufficient dynamic visualization.	Yin et al.
Machine learning	The model is trained through sample data, and the trained model is used to analyze and predict data.	High degree of automation, faster training speed.	Influenced by algorithm accuracy and quality.	Zheng et al.
QFD	Obtains the components that contribute most to requirements through requirement analysis.	Strong purpose, high excavation accuracy.	Influenced by subjective experience.	Klahn et al.
TF-IDF	Takes the frequency of keywords appearing in the text and the frequency of keywords appearing in all texts as the criteria for judging the importance of keywords.	The algorithm is simple and easy to implement.	Semantic order and context in the text are ignored.	Dorji et al.

**Table 3 entropy-24-00549-t003:** Parts of speech.

Tag	Explanation	Tag	Explanation
aux	auxiliary (be)	Root	root node
conj	conjunct	Subj	subject
nsubj	nominal subject	Obj	object
nsubjpass	passive nominal subject	Dobj	direct object
SYM	symbol	Amod	adjectival modifier
num	numeric modifier	Attr	attributive
comp	complement	punct	punctuation
…	…	…	…

**Table 4 entropy-24-00549-t004:** Types of SAO structure.

No	SAO Structure	Example
1	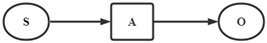	The switching mechanism is disposed on fixing holder” (US10464077):1. switching mechanism-disposed-holder.
2	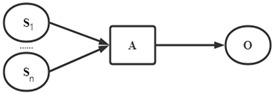	the forward position and the backward position are arranged at two sides of the initial position (US10449559):1. forward position-arranged at-initial position2. backward position-arranged at-initial position.
3	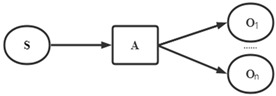	Preferably, the rotation valve comprises a sleeve, a valve core unit, a rotation support and a switch knob (US8720799):1. rotation valve-comprises-sleeve.2. rotation valve-comprises-valve core unit.3. rotation valve-comprises-rotation support.4. rotation valve-comprises-switch knob.
4	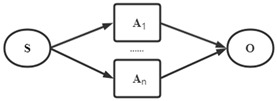	The present invention has arisen to mitigate and/or obviate the afore-described disadvantages (US20150354186A1):1. Invention-mitigate-disadvantages.2. Invention-obviate-disadvantages.
5	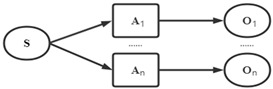	The temperature display is disposed on the curved pendant and connected with the temperature sensor (US20180202135A1):1. temperature display-disposed on-curved pendant.2. temperature display–connected- temperature sensor.

**Table 5 entropy-24-00549-t005:** Function diagram of series components.

No	Component	Functionality	Component
1	motor	drive	liquid-ctrl box
2	liquid-ctrl box	hoist	pump
3	liquid-ctrl box	pass-through	pipeline
4	motor	drive	pump
5	pump	drive	hydraulic oil
6	filter	filtration	hydraulic oil
7	pipeline	conduction	hydraulic oil
8	hydraulic-oil	drive	piston
9	cylinder	storage	hydraulic oil
10	cylinder	guide	piston
…	…	…	…

**Table 6 entropy-24-00549-t006:** US shower patent search results.

Retrieval of Patents	Result
Title or Abstract:(showerhead * OR shower head * OR sprayer *) AND CPC:(B05B1/18) AND Time:(from 1 January 1914 to 1 December 2019)	1733

**Table 7 entropy-24-00549-t007:** Patent text SAO acquisition results.

No	Patent Number	S	A	O
1	US20150273490A1			
2	US20150273490A1	Embodiments	include	use
3	US20150273490A1	Embodiments	include	showerhead
4	US20150273490A1	Embodiments	include	invention
5	US20150273490A1	Embodiments	include	chamber
6	US20150273490A1	Embodiments	include	insulator
	…	…	…	…
1048569	US10207280			
1048570	US20150273490A1	body	is	chamber
1048571	US20150273490A1	body	is	outlets
1048572	US20150273490A1	body	is	joint

**Table 8 entropy-24-00549-t008:** Component metrics.

Id	Weighted Degree
valve	6881
hole	7950
nozzle	3615
plate	5036
handle	2513
…	…
conduit	1085
cartridge	1073
controller	1051
sensor	846
shaft	2205

**Table 9 entropy-24-00549-t009:** Calculation results of CRO and CRI.

Rank by CI	Component	CRO	CPI	CI
1	valve	0.702236961	1	0.85111848
2	hole	0.587594404	0.967555	0.777574702
3	outlet	0.558008416	0.964982	0.761495208
4	plate	0.571470938	0.866782	0.719126469
5	inlet	0.527199305	0.855714	0.691456652
6	pipe	0.537497082	0.824818	0.681157541
7	axis	0.521848944	0.81001	0.665929472
8	nozzle	0.56987848	0.738293	0.65408574
9	handle	0.562902512	0.713606	0.638254256
10	connector	0.537076218	0.71216	0.624618109
11	channel	0.520952	0.71189	0.61642122
12	showerhead	0.558708	0.664609	0.611658319
13	casing	0.516216	0.664802	0.590509075
14	cavity	0.510695	0.656919	0.583807209
15	tube	0.508913	0.652626	0.58076929
16	hose	0.526853	0.625187	0.576020133
17	groove	0.502273	0.622823	0.562548247
18	ring	0.516556	0.600572	0.558564007
19	cover	0.525866	0.565686	0.545776232
20	arm	0.516927	0.555637	0.536281896
	…	…	…	…

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
