# Peer review of "Research on Product Core Component Acquisition Based on Patent Semantic Network"

_entropy, 2022, doi:10.3390/e24040549_

Round 1

Reviewer 1 Report

In the revised version, the authors do not react to the objection that only less relevant results are found. They should aim at identifying the main questions how showerheads can be improved and which main solutions were suggested.

I suggest that the authors introduce a part that their approach leads to the identification of the relevant components, but not for the identification of the main problems and the main solutions. This implies additional research to identify the parts of patent texts where these aspects are addressed. When these parts are analysed their methodology can lead to useful resultls.

Author Response

Dear Editor,

Many thanks for giving us an opportunity to revise our paper for possible publish in the Entropy. The paper has Title: Research on product core component acquisition based on patent semantic network. We have revised our paper according to your suggestions, and marked the modifications with red color. The reviewers’ comments and the modifications are described as follows.

Reviewer #1:

  1. In the revised version, the authors do not react to the objection that only less relevant results are found. They should aim at identifying the main questions how showerheads can be improved and which main solutions were suggested.

Answer: Thank you very much for your kindly comments on our manuscript. There is no doubt that these comments are valuable and very helpful for revising and improving our manuscript.

We re-examined the problems in the papers and consulted related literature, we found that the main goal of some papers is to extract component information from patent texts to identify core components, and some papers find inspection problems on the basis of identifying core components and come up with solutions, in the original paper we just belonged to the former. After listening to your suggestions, we have carried out some explorations based on existing research results, which we hope will help to improve the product.

  1. I suggest that the authors introduce a part that their approach leads to the identification of the relevant components, but not for the identification of the main problems and the main solutions. This implies additional research to identify the parts of patent texts where these aspects are addressed. When these parts are analyzed their methodology can lead to useful results.

Answer: Many thanks for your comments. We re-examined the issues in the paper and made some explorations based on the existing research results. Firstly, according to the retrieval formula in Table 6, we obtained the US shower patents from 20200101-20211231 as test dataset. And the test results shown that the weight calculation of shower components by patent data has high scores of precision value, recall value and f-value, and this can prove that the method is reliable. The designer designed the product according to the method proposed in this paper: focusing on core components will still yield better results.

For further verify the method, a new shower design scheme with electric valve is introduced based on the research result. Compare to the shower in market, the new shower has multi-way and can spray kinds of water flow for meet different customers’ requirements. The explanation and illustration are shown in page 19-20, line 551-576. The result is shown as follows:

Line 551-576:

“Furthermore, we select the us patents applicated during 202001 to 20211231 as test dataset, included 72 patents and 30,000 words, to evaluate the predictive ability of the method, and use P, R and F calculated by equation (11-13) as ability indexes. By importing data in table 9 into test dataset, the scores of P, R and F are 0.75, 0.79 and 0.77. Those high scores means that the performance of the proposed method is reliability. After all, it’s hard to detach a few valuable components from a huge amount of patents data for R&D.

According to table 7, it can find that valve is the core component of shower by patent mining. In fact, the valve of shower in market is mainly divided into single-way, double-way, or three-way. Such shower can reject fewer water flow patterns and it’s hard to meet the diverse customer needs. In reaction to this, we designed a new shower with a motor-controlled valve which can spray multiple types of water flow, and shown in figure 8. The working principle of this shower is introduced as follows: when the water through the coil 5, the coil will generate electricity, which will be stored in the battery placed inside showerhead 25, and the battery will be used to support the motor 14 to adjust the valve 15, thereby realizing water flow control. It is noted that the motor can be controlled by chip which is not shown in the figure. Compared with existing products, the valve has simpler mechanical structure and automatic adjust function by using motor and electro circuit.

Figure 8. Shower product structure

  1. inlet joint; 2. handle; 3. magnet impeller; 4. spacer casing; 5. coil; 6. sealing cover; 7. O-ring; 8. reversing stud switch; 9. screw; 10. sealing cover; 11. irregular shape ring; 12. cavity; 13. limit block; 14. motor; 15. valve; 16. long screw; 17. surface cover; 18. waterproof sealant; 19. shaft; 20. limit block; 21. gasket; 22. Water distributor connector; 23. Screw; 24. decorative cover; 25. showerhead”

Thank you again for your positive and constructive comments and suggestions on our manuscript.

Reviewer 2 Report

Some comparison tables should be added in section 2 for Methods of Patent Analyzing and Methods of Core Component Acquisitioning. The writing is not very clear, so I suggest adding some tables to explain the differences and similarities.

In section 3.1, In the concept of resources occupied in the component, it can be measured in two aspects: â…°) the quantity of adjacent components; and â…±) the quality of adjacent components. Suppose the relationship of components in a product as a matrix, the size of the adjacent component matrix depends on the number of adjacent components. Therefore, the nodes with more adjacent components have higher participation in the product, which will play a more prominent role. In fact, some components are not only adjacent with many other components but also adjacent with other highly important components. The view that only considering the quantity but ignore the quality factor is not reasonable. Thus, to decide whether a component is an essential one depends on both quality and quantity of its adjacent components. This process is not very clear. A flow chart should be provided to illustrate this.

In line 310, it could be noted that a similar method to evaluate the wellbeing scores in a structured population, see paper Online screening of X-system music playlists using an integrative wellbeing model informed by the theory of autopoiesis.

Line 379, it is mentioned Figure 4 shows that node B is the only way for other nodes to contact. According to the intermediary centrality algorithm in the structural hole theory, the two non-direct contact nodes get contact through other nodes will be controlled and restricted by the nodes on the path. In other words, node B is on the shortest path of other nodes. Consider the product as a network and the components as nodes. Suppose the shortest path number from component q to component j is qig, the number of shortest paths from element q to element j passing through element i is i qjn , then the betweenness centrality calculation of node i is shown in formula (6). The formula is not well explained. Has it been used in the same scenario? It would be better to show the network with nodes and edges. Some illustrate using gephi etc would be more attractive.

Author Response

Dear Editor,

Many thanks for giving us an opportunity to revise our paper for possible publish in the Entropy. The paper has Title: Research on product core component acquisition based on patent semantic network. We have revised our paper according to your suggestions, and marked the modifications with red color. The reviewers’ comments and the modifications are described as follows.

Reviewer #2:

  1. Some comparison tables should be added in section 2 for Methods of Patent Analyzing and Methods of Core Component Acquisitioning. The writing is not very clear, so I suggest adding some tables to explain the differences and similarities.

Answer: Many thanks for your comments. We have revisited the Literature Review section and made adjustments to the relevant studies. The literature review structure has been checked and corrected on page 3-5, line 114-116, line 124-126, line 134-142, line 146-149, line 159-163, the comparison of Methods of Patent Analyzing is shown in Table 1, the comparison of Methods of Core Component Acquisitioning is shown in Table 2. The result is shown as follows:

Line 114-116:

“Patent analysis methods are mainly divided into four categories: complex network-based analysis methods, vector-based analysis methods, TRIZ-based analysis methods, and keyword map-based analysis methods.”

Line 124-126:

“Jiang and Zhou [27], through the establishment of complex network and small-world model, explore the patent performance of industry university research cooperation;”

Line 134-142:

“Kim M, Park Y and Yoon J [31] build a patent map through patent semantic topic analysis to discover the inheritance relationship between patents. The comparison of patent analysis methods is shown in Table 1:

Table 1.  Comparison of patent analysis methods

Method

Mode of action

Advantage

Disadvantage

Literature

complex

network

The keywords in the patent text are regarded as nodes, the associations between keywords are regarded as edges, and the complex patent network is constructed to analysis

Strong visualization, which is conducive to clarifying the relationship between keywords and facilitating network analysis

Insufficient dynamic visualization

Iwan, Thorsten and Katja

vector

Perform word vector training on the domain corpus to construct technical efficacy topics

Suitable for large databases, high degree of automation

Lack of judgment on the semantic connection of keywords

Park, Chun and Jeong

TRIZ

Through the analysis and extraction of patent knowledge, it is introduced into TRIZ tool to provide a large number of heuristic principles, effects, structures, etc. for solving product innovation problems in specific fields.

not only makes up for the limitations of TRIZ and the ambiguity and broadness of the obtained solutions, but also makes up for the microscopic nature of knowledge acquired through patents.

Relying on the designer's subjective experience and domain knowledge

Li et al

keyword map

Transform technical information in patents into a map of technology-directed functionality

The content is detailed and helpful for understanding technological trends

difficult to find and organize information.

Kim M, Park Y and Yoon J

After comparative analysis, the patent analysis method based on complex network considers the semantic information in the patent text, it can be used to realize network analysis, and has better visualization effect, so it has gradually become a popular method for patent analysis.”

Line 146-149:

“The identification methods of core components are mainly divided into four categories: based on network algorithm, based on machine learning, based on Quality Function Deployment (QFD) and based on Term Frequency–Inverse Document Frequency (TF-IDF).”

Line 159-163:

“Dorji et al. [36] applied TF-IDF to domain-related terms to realize the identification and extraction of domain-related terms. The comparison of core component identification methods is shown in Table 2:

Table 2.  Comparison of core component identification methods

Method

Mode of action

Advantage

Disadvantage

Literature

network algorithm

Map product components to networks, and identify core components through network measurement algorithms.

easy to measure the position and use of nodes in  network, Strong visualization.

Insufficient dynamic visualization.

Yin et al

machine learning

The model is trained through sample data, and finally the trained model is used to analyze and predict the data.

High degree of automation, faster training speed.

Influenced by algorithm accuracy and quality.

Zheng et al

QFD

Obtain the components that contribute the most to requirements through requirements analysis.

Strong purpose, high excavation accuracy.

Influenced by subjective experience.

Klahn et al

TF-IDF

Take the frequency of keywords appearing in the text and the frequency of keywords appearing in all texts as the criteria for judging the importance of keywords

The algorithm is simple and easy to implement.

Semantic order and context in the text are ignored.

Dorji et al

  1. In section 3.1, In the concept of resources occupied in the component, it can be measured in two aspects: â…°) the quantity of adjacent components; and â…±) the quality of adjacent components. Suppose the relationship of components in a product as a matrix, the size of the adjacent component matrix depends on the number of adjacent components. Therefore, the nodes with more adjacent components have higher participation in the product, which will play a more prominent role. In fact, some components are not only adjacent with many other components but also adjacent with other highly important components. The view that only considering the quantity but ignore the quality factor is not reasonable. Thus, to decide whether a component is an essential one depends on both quality and quantity of its adjacent components. This process is not very clear. A flow chart should be provided to illustrate this.

Answer: Many thanks for your comments. We thought the reviewers wanted more elaboration on the definition. In this regard, we will introduce the definition in detail through the case of a shower. The flow chart of the process to measure resources occupied in the component in Figure 1 on page 6, line 194-207. The result is shown as follows:

Line 194-207:

“In the concept of resources occupied in the component, it can be measured in two aspects: i) the quantity of adjacent components; and ii) the quality of adjacent components. Suppose the relationship of components in a product as a matrix, the size of the adjacent component matrix depends on the number of adjacent components. Therefore, the nodes with more adjacent components have higher participation in the product, which will play a more prominent role. In fact, some components are not only adjacent with many other components but also adjacent with other highly important components. The view that only considering the quantity but ignore the quality factor is not reasonable. Thus, to decide whether a component is an essential one depends on both quality and quantity of its adjacent components. For example, the shower is composed by 10 components which is shown in figure 1. The adjacent matrix shows that the component 3 and 4 have are connect with same number components, but the weight of component 3 (W3=8) is larger than component 4 (W4=7).

Figure 1.  Example of core component

  1. In line 310, it could be noted that a similar method to evaluate the wellbeing scores in a structured population, see paper Online screening of X-system music playlists using an integrative wellbeing model informed by the theory of autopoiesis.

Answer: Many thanks for your comments. We have carefully read this literature and cite it in this paper to demonstrate the usability of the scheme on page 11, line 332-333. The result is shown as follows:

“Since β1 is the most apparent basic interpretation, the weight of β1 is defined at above 0.5 according to the research of [46].”

  1. Line 379, it is mentioned Figure 4 shows that node B is the only way for other nodes to contact. According to the intermediary centrality algorithm in the structural hole theory, the two non-direct contact nodes get contact through other nodes will be controlled and restricted by the nodes on the path. In other words, node B is on the shortest path of other nodes. Consider the product as a network and the components as nodes. Suppose the shortest path number from component q to component j is qig, the number of shortest paths from element q to element j passing through element i is i qjn , then the betweenness centrality calculation of node i is shown in formula (6). The formula is not well explained. Has it been used in the same scenario? It would be better to show the network with nodes and edges. Some illustrate using gephi etc would be more attractive.

Answer: Many thanks for your comments. We have repainted the Figure of “Example of structure hole network” using Gephi, the Figure is shown on page 13, line 383-384. The situation in this figure is used as a case to explain Formula 6 on page 13, line 410-415. The result is shown as follows:

Line 383-384:

Figure 5.  Example of structure hole network

Line 410-415:

“Taking the situation in Figure 5 as an example, the shortest paths between non-directly connected nodes are: A-B-D-E, A-D-E, A-C-D-E, B-D-C, B-A-C, B-D-E, C-D-E. That is,, , , , the shortest paths from node A to node E all pass through D, then , and only one shortest path from node B to node C passes through node D, so , and so on,  , . finally, the betweenness centrality of the node D is  .”

Thank you again for your positive and constructive comments and suggestions on our manuscript.

In addition, in the process of revising our paper, we have found that the sorting of the charts was not accurate, we have corrected this.

Other misspellings, oral expressions and grammar mistakes have been checked and corrected by native English-speaking professionals. The modified places will not be described in detail here.

Round 2

Reviewer 1 Report

The paper can be published n the present form. However, a critcal assessment as to the focus of inventions of the last years is still missing.

Reviewer 2 Report

I have no further comment. The paper can be accepted. 

This manuscript is a resubmission of an earlier submission. The following is a list of the peer review reports and author responses from that submission.

Round 1

Reviewer 1 Report

The literature review structure needs to be adjusted. Currently, it starts with patent analysis based on text mining and patent analysis based on complex network and then you conclude it in a summary. The structure of the two patent analysis is not clear and often muddy. They are not parallel. You should have provided a systematical literature review and created a streamline of the results in terms of topics or in terms of chronicle order. 

The motivation and contribution should be highlighted in the introduction. It would be useful if you could provide a flow chart to show the progress of the ideas and set out the scheme.

A time complexity analysis would be critical for the algorithm of SAO extraction process. This is essential for the type of study. The effectiveness should be solid.

It is mentioned that Sememe is the smallest unit of interpretation, and a finite set of sememes can express an interpretation. When calculating sememe similarity, it is usually calculated by the path length between two sememes in the sememe tree. When two sememes are on different sememe trees, it is considered that the path length between two sememe nodes is infinite, which means the similarity between them is 0. To calculate the interpretation
similarity, formula (3) can be applied. This is not clear to the reviewer. The equation should be explained.

The component importance is interesting. What is the rationality of this formula?

Figure 6 is very blurred. A figure with higher resolution should be provided in the revision.

The language of this paper needs to be improved significantly. I suggest a professional editing service if available.

Author Response

Response to Reviewer 1 Comments

Point 1: The literature review structure needs to be adjusted. Currently, it starts with patent analysis based on text mining and patent analysis based on complex network and then you conclude it in a summary. The structure of the two-patent analysis is not clear and often muddy. They are not parallel. You should have provided a systematical literature review and created a streamline of the results in terms of topics or in terms of chronicle order.

Response 1: Many thanks for your comments. This is a mistake, and the literature review structure has been checked and corrected on page 2-4, line 77-112, line 152-156. The structure has been divided as four parts: Methods of Patent Information Mining, Methods of Patent Analyzing, Methods of Core Component Acquisitioning and Summary. The result is shown as follows:

“It is mainly divided into three categories for the method of patent text information mining: patent information mining based on domain knowledge [14], patent information mining based on vector space [15] and patent information mining based on pa-tent text semantics [16]. The patent information mining based on domain knowledge can comprehensively mine patent technology information by constructing domain knowledge ontology, the information in the patent text is represented by the ontology. However, the construction of domain knowledge requires a lot of manual participation, which is time-consuming and labor-intensive [17]. Patent information mining based on vector space models patent information as a vector, to explore the spatial relationship between vectors, which has the advantage of mature technology [18]. The patent in-formation mining based on the semantic information of the patent text represents the text with semantics as the association rules, represents the patent information through semantic mining and semantic relationship mapping, to realize the full mining of the deep patent information, thus it has gradually become a research direction favored by the academia [19].

For the purpose of make the semantic information of patent texts more sufficient, and the technical features of patents to be expressed more clearly, scholars use more efficient methods to mine the semantic information of patent texts, such as entity extraction and SAO. Among them, Ki and Kim[20] used SAO to extract the technical information in the patent by summarizing the defects of the existing analysis methods, so as to reduce the manual operation as much as possible when extracting the technical description content, and display the results in the form of a matrix; As for An et al.[21], in order to determine the relationship between keywords in patents, prepositions are introduced into the se-mantic analysis network, thereby overcoming the limitations of keyword network analysis; Wang et al.[22] considering the relationship between different SAOs when using SAO to mark patent texts, the DWSAO framework is proposed to assign different weights to different SAOs, which improves the extraction efficiency of patent text information. In addition to SAO, Chen et al.[23] built a pa-tent information extraction framework through named entity recognition and semantic relation extraction, which contributed to the performance improvement of named entity extraction in patent texts; Xiong et al.[24] monitoring, identifying and extracting named entities in drug patents, and matching knowledge graphs, increasing the efficiency of drug information mining in patents to 83%, providing reliable data for building drug monitoring programs. Compared with SAO, named entity recognition method extracts less deep text information, and cannot relate between words and words in patents [25].”

“Judging from the research status of component identification, patent information mining and patent analysis methods, the existing research mostly focuses on mining product technical information in patent texts, so as to achieve technical efficacy analysis, cluster analysis, etc. but little research has been done in identifying and acquiring core components [36-42].”

Point 2: The motivation and contribution should be highlighted in the introduction. It would be useful if you could provide a flow chart to show the progress of the ideas and set out the scheme.

Response 2: Many thanks for your comments. The motivation and contribution have been highlighted in introduction on page 1-2, line 44-49. The result is shown as follows:

“From the characteristics of the patent text, it can be found that the patent text contains rich product information. In order to fully explore product structure information from patent texts and eliminate doubts for product designers, this study pro-poses a core component identification method based on patent texts, which combines the requirements for core component acquisition and the technical advantages of patent text analysis.”

Point 3: A time complexity analysis would be critical for the algorithm of SAO extraction process. This is essential for the type of study. The effectiveness should be solid.

Response 3: Many thanks for your comments. This is a mistake not considered. The time complexity analysis and result has been illustrated on page 9, line 279-280. The result is shown as follows:

“Since the algorithm contains two “for” loops, the time complexity of the algorithm T(n) = O(n2).”

Point 4: It is mentioned that Sememe is the smallest unit of interpretation, and a finite set of sememes can express an interpretation. When calculating sememe similarity, it is usually calculated by the path length between two sememes in the sememe tree. When two sememes are on different sememe trees, it is considered that the path length between two sememe nodes is infinite, which means the similarity between them is 0. To calculate the interpretation similarity, formula (3) can be applied. This is not clear to the reviewer. The equation should be explained.

Response 4: Many thanks for your comments. The explanation of formula (3) has been added and an example from Figure 2 has been given on page 10, line 322-326. The result is shown as follows:

“Taking the sememe tree in Figure 2 as an example, "Root" is the root node, and other nodes can find at least one path to reach the root node. In addition, B1 is the root node of B11 and B12, and so on. When two nodes do not have a common root node, it means that the two nodes have no path to reach each other, so it can be regarded that the meanings expressed by the two sememes are not similar.”

Point 5: The component importance is interesting. What is the rationality of this formula?

Response 5: Many thanks for your comments. In former research, the identification and acquisition of component importance has not considered the location and association of component in product. Formula (10) can make up for the shortcomings of existing research. Among them, CPI is based on the location of components, and CRO is based on the resources (association) owned by components. Through parameter adjustment, component importance data under different focuses can be obtained. These are described in Section 4.3.1 and Section 4.3.2. In response to the rationality of this formula, the explanation has been modified on page 13, line 436-441. The result is shown as follows:

       “In former research, the identification and acquisition of component importance has not considered the location and association of component in product. Formula (10) can make up for the shortcomings of existing research. Among them, CPI is based on the location of components, and CRO is based on the resources (association) owned by components. Through parameter adjustment, component importance data under different focuses can be obtained.”

Point 6: Figure 6 is very blurred. A figure with higher resolution should be provided in the revision.

Response 6: Many thanks for your comments. The complex network was shown in a network with nodes and edges, the radius of nodes presents the size of nodes’ value, the width of edges presents the size of the association strength between nodes. In order to show which nodes are relatively important, the colors of nodes and edges are also defined, so the figure seems blurred. it's regrettable.

Point 7: The language of this paper needs to be improved significantly. I suggest a professional editing service if available.

Response 7: Many thanks for your comments. We have carefully checked and improved the English writing in the revised manuscript.

In addition, in the process of revising our paper, we have found that the sorting of the charts was not accurate, we have corrected this. 

       Thank you again for your positive and constructive comments and suggestions on our manuscript.

Reviewer 2 Report

The paper deals with the extraction of technical information from patent documents by text analysis. In particular, it uses natural language processing (NPL), part of speech (POS) tagging and sub-subject-action-object (SAO). Firstly, to patent’s keywords from the products are extracted, then complex network are applied to obtain core components based on structural holes and centrality of eigenvector algorism. This methodology is illustrated by the example of showerheads.

- The methodology is very sumptuous and laborious, but in the end, the results are quite sobering. A list of components of a device, e.g. parts of a showerhead is not interesting. Every company, producing showerheads is well aware of the parts of a showerhead and does not need this information.

- It would be interesting to achieve information of the innovation activities of the last ten years. What were the major problems that were addressed. What were the major solutions? Which components were modified? Which components were omitted? Which new components were introduced. For this purpose, other parts of the patent documents have to be analysed, and different methods have to be used.

- For newcomers, it is sufficient to read manuals of showerheads, books od technical schools, two or three patent documents or simply buy a showerhead in an standard store to get knowledge about key components.

- Besides, the CPC has about 250000 items. Two items for a simple technology such as showerheads is sufficient B05B1/18 and /185.

Author Response

Response to Reviewer 2 Comments

Reviewer #2: The paper deals with the extraction of technical information from patent documents by text analysis. In particular, it uses natural language processing (NPL), part of speech (POS) tagging and sub-subject-action-object (SAO). Firstly, to patent’s keywords from the products are extracted, then complex network are applied to obtain core components based on structural holes and centrality of eigenvector algorism. This methodology is illustrated by the example of showerheads.

Point 1: The methodology is very sumptuous and laborious, but in the end, the results are quite sobering. A list of components of a device, e.g. parts of a showerhead is not interesting. Every company, producing showerheads is well aware of the parts of a showerhead and does not need this information.

Response 1: Many thanks for your comments. Firstly, as the comments say, it’s a fact that every company is aware of the parts of a showerhead, but the knowledge of showerhead’s component depends on the designers’ experience, designer’s subjectivity affects design results. Moreover, if the result of this research seems consistent with the designer's experience, that means our research doesn't contradict the facts. Finally, compared to the designer's experience, the results of this research seem more objective, and there are calculation results as support to make the results more intuitive. Thanks again for your pertinent comment, it made us take this into consideration when choosing our subjects.

Point 2: It would be interesting to achieve information of the innovation activities of the last ten years. What were the major problems that were addressed. What were the major solutions? Which components were modified? Which components were omitted? Which new components were introduced. For this purpose, other parts of the patent documents have to be analysed, and different methods have to be used.

Response 2: Many thanks for your comments. We realize the comments meant to suggest that we work on trend analysis, technology opportunity discovery. It is believed that some interesting phenomena will be found through the analysis of the patent text and its structured data. These suggestions have provided us with good ideas and pointed out the direction for our next research. Thanks again.

Point 3: For newcomers, it is sufficient to read manuals of showerheads, books on technical schools, two or three patent documents or simply buy a showerhead in an standard store to get knowledge about key components.

Response 3: Many thanks for your comments. Shower patents or manuals can indeed help newcomers understand the structure and characteristics of the product, but they are instructions for a specific product and have certain limitations. AS for buy the product and dismantle it, the disadvantages have been introduced on section 1, line 12: “inefficient and costly”. There are so many different products on the market, we need to find the common features in the products and find the most noteworthy parts in the form of data. Thus, make full use of patent data enables to gain an objective understanding of the components in the product. Thanks for your valuable counsel.

Point 4: Besides, the CPC has about 250000 items. Two items for a simple technology such as showerheads is sufficient B05B1/18 and /185.

Response 4: Thank you for your significant reminding. Based on our experimental subjects, we determined the scope of the retrieved patents, our search scope is basically the same as the comment, which are reflected in Table 4:

Table 4The U.S. shower patent search results

Retrieval of patent

Result

Title or Abstract:(showerhead* OR shower head* OR sprayer*) AND CPC:(B05B1/18) AND Time:(from 19140101 to 20200101)

1733

In addition, in the process of revising our paper, we have found that the sorting of the charts was not accurate, we have corrected this. And all the verified text are label with red color.

Thank you again for your positive and constructive comments and suggestions on our manuscript.

Round 2

Reviewer 1 Report

My questions have been addressed. I recommend acceptance.

Reviewer 2 Report

The arguments of the authors are not really convincing. The detection of relevant aspects by text analysis is challenging but nort solved in this paper.